# Imitation Learning with Human Eye Gaze via Multi-Objective Prediction

**Ravi Kumar Thakur** [*1]  **MD-Nazmus Samin Sunbeam** [*1]  **Vinicius G. Goecks** [2]  **Ellen Novoseller** [2]  **Ritwik Bera** [1]
**Vernon J. Lawhern** [2]  **Gregory M. Gremillion** [2]  **John Valasek** [1]  **Nicholas R. Waytowich** [2]

## Abstract

Approaches for teaching learning agents via human demonstrations have been widely studied and successfully applied to multiple domains. However, the majority of imitation learning work utilizes only behavioral information from the demonstrator, i.e. which actions were taken, and ignores other useful information. In particular, eye gaze information can give valuable insight towards where the demonstrator is allocating visual attention, and holds the potential to improve agent performance and generalization. In this work, we propose Gaze Regularized Imitation Learning (GRIL), a novel context-aware, imitation learning architecture that learns concurrently from both human demonstrations and eye gaze to solve tasks where visual attention provides important context. We apply GRIL to a visual navigation task, in which an unmanned quadrotor is trained to search for and navigate to a target vehicle in a photo-realistic simulated environment. We show that GRIL outperforms several state-of-the-art gaze-based imitation learning algorithms, simultaneously learns to predict human visual attention, and generalizes to scenarios not present in the training data. Supplemental videos and code can be found at `https://sites.google.com/view/gaze-regularized-il/`.

## 1. Introduction

In the human-robot interaction field, imitation learning (IL), also called learning from demonstration, is widely used to rapidly train artificial agents to mimic the demonstrator via supervised learning (Argall et al., 2009; Osa et al.,

*Equal contribution [1]Department of Aerospace Engineering, Texas A&M University, College Station, Texas, USA [2]DEVCOM Army Research Laboratory, Aberdeen Proving Ground, Maryland, USA. Correspondence to: Ravi Kumar Thakur <ravikt@tamu.edu>, MD-Nazmus Samin Sunbeam <mdsunbeam@tamu.edu>.

Interactive Learning with Implicit Human Feedback Workshop at ICML 2023.

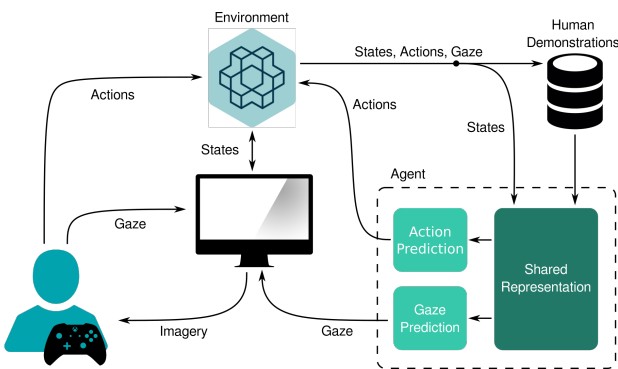

*Figure 1.* **GRIL system diagram:** The proposed multi-objective learning architecture learns from human demonstrations consisting of actions and eye gaze data to train an imitation learning policy.

2018). IL using human-generated data has been widely studied and successfully applied to multiple domains such as self-driving cars (Codevilla et al., 2019), robot manipulation (Rahmatizadeh et al., 2018), and navigation (Silver et al., 2010). Yet, while IL is a simple and straightforward approach for teaching intelligent behavior, it suffers from sample complexity issues when learning an end-to-end behavior policy directly from images, e.g. mapping images from a robot's camera to actions.

One avenue to improving IL sample efficiency and generalizability is to learn not only from the demonstrator's actions, but also from additional signals such eye gaze. While the majority of existing work on IL ignores physiological data (Argall et al., 2009), eye gaze is a rich signal that has been shown to strongly correlate with visual attention (Doshi & Trivedi, 2012), guide our actions, and filter parts of the environment perceived as relevant (Schütz et al., 2011). Thus, eye gaze may provide important contextual information about a person's thought process and provide an indication of visual attention that can be leveraged when training AI agents. For example, when people provide demonstrations for tasks such as flying a quadrotor via a joystick, eye gaze is often disregarded or ignored, even though it is necessary to perform the task and can be easily collected via widely available eye tracking hardware (Poole & Ball, 2006) at almost no additional burden to the demonstrator. In our work, we leverage the demonstrator's eye gaze, recorded

with minimal cost during the demonstrations, as a measure of attention in conjunction with human demonstrations to perform imitation learning with less demonstrator data.

Several previous works have utilized eye gaze to improve imitation learning (Zhang et al., 2018; Saran et al., 2020; Chen et al., 2019). However, these methods may still require significant human demonstrator time. For instance, the gaze-augmented Atari-HEAD dataset (Zhang et al., 2020) collects an average of 5.85 hours of gameplay data per Atari game. Furthermore, the applications considered in these works may be significantly simpler than the environment a real-world robot might encounter. For instance, the Atari setup in Zhang et al. (2018; 2020); Saran et al. (2020); Thammineni et al. (2021) is synchronous and does not resemble the physical world, the simulated driving setup in Chen et al. (2019) involves following a well-defined track, and the drone task in Pfeiffer et al. (2022) involves following a reference trajectory (provided to the policy network) without interacting with any objects in the environment.

We propose Gaze-Regularized Imitation Learning (GRIL), a novel end-to-end algorithm for learning continuous control from human demonstrations and eye gaze. GRIL jointly learns to predict control commands and eye gaze, regularizing policy training via gaze prediction as shown in Figure 1. This method has the potential to improve robust autonomy and decrease the amount of demonstration data required to learn reliable policies in visually complex environments.

We demonstrate GRIL in an autonomous quadrotor control task in the AirSim (Shah et al., 2018b) simulator, training a visuomotor policy (i.e., combining perception and control) to navigate a quadrotor to a target object. GRIL is trained in supervised fashion via a dataset collected from human demonstrators flying the quadrotor in the simulator, which contains the RGB images, human demonstrator's eye gaze coordinates, and control commands at each timestep. The proposed method jointly learns to predict gaze coordinates and control commands. In the quadrotor navigation task, we hypothesize that the gaze coordinates provide context relevant to the target, identifying regions of interest in the scene and acting as an attention mechanism to guide the policy toward the target. Specifically, we contribute:

1. GRIL, an end-to-end, model-free algorithm with a novel multi-objective architecture for learning from images, demonstrator actions, and human eye gaze. To our knowledge, GRIL is the first IL method to regularize policy learning via a multi-objective architecture that jointly predicts gaze and policy actions.

2. A demonstration of our gaze-based approach using an asynchronous, realistic quadrotor navigation task with high-dimensional state and action spaces and a high-fidelity, photorealistic simulator. To our knowledge, this is the most complex task considered in a

gaze-augmented imitation learning work, as it requires continuous control in a photorealistic, outdoor environment without clear markers such as roads denoting where the agent should travel.

3. A quantitative evaluation showing that GRIL is able to significantly outperform three baseline methods: a standard behavior cloning model that does not use gaze as well as two state-of-the-art gaze-augmented behavior cloning models: AGIL (Zhang et al., 2018) and CGL (Saran et al., 2020)

4. We show that after training with demonstrations in which the quadrotor navigates to a stationary target, GRIL generalizes to a novel "seek and follow" task in which the quadrotor must navigate to and follow a moving target.

5. We provide the first publicly-available gaze-augmented demonstration dataset in a continuous control task.

## 2. Related work

**Imitation Learning.** Imitation learning (IL) is the problem of training a learning agent to act in an environment given demonstrations. Behavioral cloning (Bratko et al., 1995; Torabi et al., 2018) is a popular IL method that uses supervised learning to predict demonstrator actions given observations. Behavioral cloning is known to suffer from covariate shift, in which a model gives poor predictions in states not present in the training data (Ross et al., 2011). In recent years, a number of other IL approaches have been proposed, for instance GAIL (Ho & Ermon, 2016), IQ-learn (Garg et al., 2021), SQIL (Reddy et al., 2019), ValueDICE (Kostrikov et al., 2019), Maximum Likelihood IRL (Jain et al., 2019), EDM (Jarrett et al., 2020), and T-REX (Brown et al., 2019). We note that advancements in IL are orthogonal to this work, since our gaze regularizer can be paired with any IL method.

**Understanding Gaze Behavior.** Recent works have proposed a number of methods for estimating human eye gaze from images (Li et al., 2013; Xia et al., 2020; Wang & Sung, 2002; Cazzato et al., 2020). For instance, Xia et al. (2020) proposed a periphery-fovea multi-resolution model to predict gaze and show that it improves prediction accuracy in a supervised learning task in a driving domain. While these works focus on modeling gaze, our work leverages gaze to improve performance in IL tasks.

Saran et al. (2019) highlight the importance of understanding gaze behavior for use in IL. For several goal-oriented robot manipulation tasks, the authors show that users primarily fixate on goal-related objects and that gaze can help to resolve ambiguities in subtask classification. Meanwhile, Guo et al. (2021) demonstrate that reinforcement learning agents achieve better performance when they attend to simi-

lar visual targets to humans, further indicating that human eye gaze can be leveraged to guide policy learning.

**Gaze-Augmented Imitation Learning.** IL approaches that augment demonstrations with the demonstrator's eye gaze have been shown to improve policy performance and generalization compared to IL methods without gaze (Zhang et al., 2018; 2020; Saran et al., 2019; 2020; Thammineni et al., 2021; Liu et al., 2021; Kim et al., 2020; 2021; 2022; Pfeiffer et al., 2022; Chen et al., 2019) in domains such as Atari, simulated driving, and robot manipulation.

These works propose a range of approaches for leveraging gaze in IL. Several methods use predicted or actual gaze to *modulate* either the control policy network or its input. For instance, some approaches that crop an image observation around a user's gaze (Kim et al., 2020; 2021; 2022) have shown promise in robot manipulation tasks requiring precision. Liu et al. (2021) similarly propose to directly modulate input images based on gaze. Liu et al. (2021) and Chen et al. (2019) consider using gaze to control dropout rates in the policy network's convolution layers. Evaluated on a simulated driving task, Liu et al. (2021) show that both the image modulation and dropout methods reduced generalization error compared to IL without gaze, with dropout outperforming the image modulation method. Meanwhile, Saran et al. (2019) also leverages gaze in robot manipulation and proposes an inverse reinforcement learning approach to learn rewards from demonstrations with gaze. However, the method requires knowing the positions of all objects of interest to which a human might attend, and thus does not necessarily generalize across domains.

A number of works (Zhang et al., 2018; Thammineni et al., 2021; Pfeiffer et al., 2022; Xia et al., 2020) estimate gaze from image observations and then leverage the gaze predictions as a control policy input. Zhang et al. (2018) develop Attention Guided Imitation Learning (AGIL), which trains a gaze prediction network modeled as a human-like foveation system and then uses the gaze predictions to train a policy. Evaluated in Atari domains, AGIL outperforms a baseline without gaze. However, to eliminate the effect of human reaction time and fatigue, eye gaze was collected in a synchronous fashion in which the environment only advanced to the next state once the human took an action, and game time was limited to 15 minutes, followed by a 20-minute rest period; this highlights the challenges of real-time human data collection. Thammineni et al. (2021) build on AGIL by introducing a gating model that selectively passes gaze information to the policy network when it predicts that gaze will be useful. While this method is shown to outperform AGIL, the contribution is orthogonal to ours, as such a gating module could be straightforwardly paired with GRIL. Unlike these works, GRIL incorporates gaze as an auxiliary loss, rather than as a policy input.

Saran et al. (2020) propose a coverage-based gaze loss (CGL) for IL. This auxiliary loss penalizes the policy network for having low network activations in areas where the human gaze is focused. CGL outperforms AGIL (Zhang et al., 2018) and dropout-based modulation (Chen et al., 2019) with the Atari-HEAD (Zhang et al., 2020) dataset. While GRIL and CGL both regularize policy learning via auxiliary losses, they are fundamentally different; whereas CGL penalizes network activations, which requires setting hyperparameters that transform the human gaze from coordinates to a heatmap, GRIL utilizes gaze directly by training a two-headed network to jointly predict actions and gaze. Moreover, GRIL's approach to utilizing gaze is distinct from CGL and AGIL in that both baselines predict and utilize a gaze heatmap, while GRIL does not utilize a heatmap. Bypassing the need for a heatmap is desirable, since GRIL does not need to specify a kernel hyperparameter and can use an MSE loss to compare coordinates, which is simpler than using a KL loss to compare distributions.

We believe that our work considers a more challenging environment than these previous works on gaze-augmented IL. Many works, including state-of-the-art approaches such AGIL (Zhang et al., 2018) and CGL (Saran et al., 2020), are exclusively tested in Atari domains, which can easily be stopped and started to synchronously align with human input, are temporally discrete, have discrete action spaces, and do not resemble the real world. In contrast, we consider a quadrotor navigation application that requires continuous control and complex scene understanding. Some works consider car driving tasks (Chen et al., 2019; Xia et al., 2020), which also require continuous control; however, our quadrotor control task is more complex than the driving tasks in Chen et al. (2019); Xia et al. (2020), as their action spaces have fewer degrees of freedom. While Pfeiffer et al. (2022) also consider drone control, they train the drone to follow reference trajectories such as a figure-8, which does not involve interacting with other objects in the scene. Also, in addition to a gaze estimate, the policy network in Pfeiffer et al. (2022) is given the reference trajectory and drone state (rotation and linear and angular velocity) as inputs. GRIL does not utilize such information.

Lastly, while the Atari-HEAD dataset (Zhang et al., 2020) provides a set of gaze-augmented demonstrations in Atari domains, we are not aware of a prior publicly-available dataset of gaze-augmented demonstrations in a continuous control setting.

## 3. Gaze-Regularized Imitation Learning

This work considers a gaze-augmented visual imitation learning setting, in which a learning agent aims to imitate a human demonstrator given the human's visual observations, actions, and eye gaze.

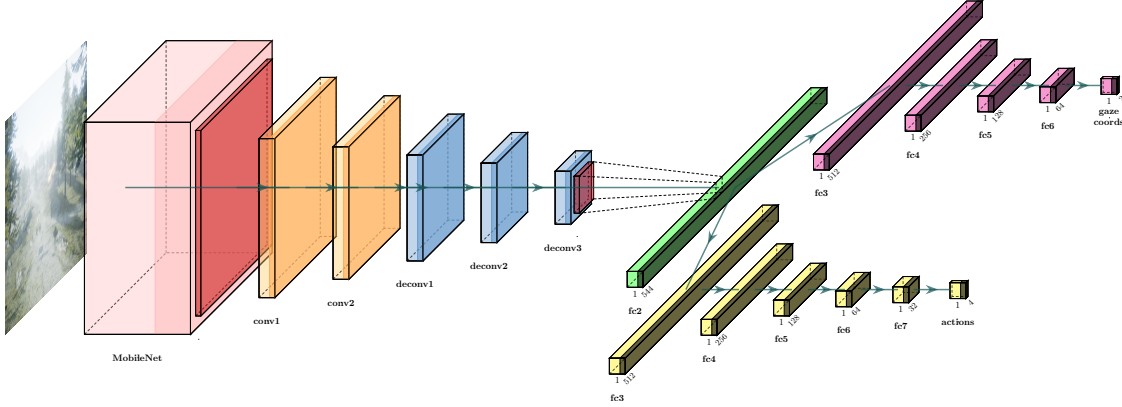

*Figure 2.* GRIL learns to jointly predict gaze and control commands via a multi-headed convolutional neural network. Gaze prediction provides context relevant to the visual scene and assists the model to predict well-performing control commands.

### 3.1. Problem Statement

We model the gaze-augmented visual imitation learning problem as an episodic partially-observable Markov decision process without reward (POMDP\R), $\mathcal{M} = (\mathcal{S}, \mathcal{O}, \mathcal{A}, P, P_e, \mu, T)$, where $\mathcal{S}$ is the underlying state space, $\mathcal{O}$ is the observation space, $\mathcal{A}$ is the action space, $P : \mathcal{S} \times \mathcal{A} \times \mathcal{S} \rightarrow [0, 1]$ yields the state transition probabilities, $P_e : \mathcal{S} \times \mathcal{O} \rightarrow [0, 1]$ gives the observation emission probabilities, $\mu : \mathcal{S} \rightarrow [0, 1]$ gives the initial state probabilities, and $T$ is the time horizon. A *policy* is a possibly-stochastic mapping from observations to actions, $\pi : \mathcal{O} \rightarrow \mathcal{A}$. The learning agent interacts with the environment in rollout trajectories of the form $\tau = (o_1, a_1, \ldots, o_T, a_T, o_{T+1})$.

We assume that the observations $o \in \mathcal{O}$ are images in $\mathbb{R}^{H \times W \times C}$, with height $H$, width $W$, and number of channels $C$ (e.g. $C = 3$ with RGB images). Furthermore, the learning agent has access to a set of gaze-augmented demonstrations consisting of $M$ observation-action-gaze triples: $\mathcal{D}_e = (o_i, a_i, g_i)_{i=1}^M$, where $o_i \in \mathcal{O}$, $a_i \in \mathcal{A}$, and $g_i \in \mathbb{R}^2$ are the demonstrator's gaze coordinates. The demonstration data is assumed to be generated by an unknown human demonstrator policy $\pi_h$.

**Learning Objective.** The imitation learning objective is to identify an optimal policy $\pi^*$ with minimal discrepancy from the demonstrator policy $\pi_h$:

$$\pi^* = \operatorname{argmin}_\pi \sum_{t=1}^T \mathbb{E}_{o \sim d_t^\pi} [\mathcal{L}(\pi_h(o), \pi(o)))], \quad (1)$$

where $d_t^\pi$ is the distribution over observations at timestep $t$ induced by following policy $\pi$, and $\mathcal{L}$ is a discrepancy measure between two actions (e.g. MSE error).

### 3.2. Gaze-Regularized Architecture

Our proposed approach is a visuomotor policy based on a multi-headed convolutional neural network, as seen in Figure 2. The model is trained to take RGB images as input and to jointly estimate their corresponding control commands and gaze coordinates. This joint estimation of gaze and control regularizes the network during training to predict the most suitable set of control commands. The gaze estimation also provides context to the model by highlighting key properties of the image observations. The architecture makes use of weight sharing to reduce the total number of model parameters.

We leverage a pre-trained MobileNet (Sandler et al., 2018) to extract features from the images; while the MobileNet weights are used to initialize our model, we further fine-tune these weights using the human demonstrations. MobileNet is trained on real-world images and is suitable for processing outdoor images. We remove the model's final layer and feed its output features through another set of convolution layers. The features extracted from the image are passed through two different sets of dense layers, which respectively predict control commands and gaze coordinates.

### 3.3. Training Criterion

We train the GRIL model via a linear combination of a *gaze prediction loss*, $\mathcal{L}_{GP}$, and a *behavioral cloning loss*, $\mathcal{L}_{BC}$:

$$\mathcal{L}(\theta) = \lambda_1 * \mathcal{L}_{GP}(\theta) + \lambda_2 * \mathcal{L}_{BC}(\theta), \quad (2)$$

where $\theta$ represents the set of trainable model parameters, and $\lambda_1$ and $\lambda_2$ are hyperparameters weighting the loss components. Each loss term is the mean-squared error between

the corresponding ground truth and predicted values:

$$\mathcal{L}_{BC}(\theta) = \frac{1}{M} \sum_{i=1}^{M} \left\| \pi_{\text{action}}(o_i \mid \theta) - a_i \right\|^2, \qquad (3)$$

$$\mathcal{L}_{GP}(\theta) = \frac{1}{M} \sum_{i=1}^{M} \left\| \pi_{\text{gaze}}(o_i \mid \theta) - g_i \right\|^2, \qquad (4)$$

where $M$ is the number of training samples, $\pi_{\text{action}}(o \mid \theta)$ and $\pi_{\text{gaze}}(o \mid \theta)$ respectively denote the outputs of the action and gaze prediction heads given observation $o$ and model parameters $\theta$, and recall that $\mathcal{D}_e = (o_i, a_i, g_i)_{i=1}^{M}$ is the demonstration dataset.

# 4. Experiments

## 4.1. Autonomous Drone Navigation

We evaluate the performance of GRIL in an autonomous quadrotor navigation domain in which experiments were conducted in simulated environments rendered by the Unreal Engine using Microsoft AirSim (Shah et al., 2018a), a high-fidelity, photo-realistic drone simulator. We consider a search and navigate task in which the quadrotor must seek a target vehicle that is initially out-of-view and navigate toward it in a cluttered forest simulation environment, seen in Figure 3. The environment emulates sun glare and presents trees and uneven rocky terrain as obstacles to navigation and visual identification of the target vehicle. We consider two variants of the drone navigation task:

1. Stationary target: the target truck is in a location that is fixed across episodes, and the target does not move.
2. Moving target: we use this task to evaluate how well GRIL can generalize to a target tracking and following task on which it was not previously trained. In this task, the target vehicle moves along a fixed, predetermined path at a fixed speed. This task requires not only seeking out the target vehicle, but also tracking and following the vehicle.

## 4.2. Dataset Collection Procedure

While performing the task, the demonstrator was given access to the quadrotor's first-person view, with no additional information about its location or the target location. Using an Xbox One joystick, the demonstrator controlled the quadrotor throttle and yaw rate using the left joystick and controlled its forward and lateral velocity via the right joystick, as is standard for aerial vehicles. The eye gaze data collection was conducted using a screen-mounted eye tracker, which was calibrated specifically for the user. Appendix A describes the eye tracking hardware setup, data collection, and calibration procedure.

We define a set of 24 possible quadrotor starting locations,

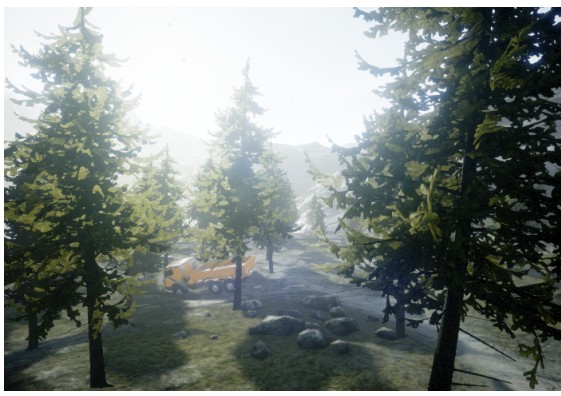

Figure 3. First-person view from the quadrotor illustrating the target vehicle (yellow truck), which the agent must find and navigate toward, and the realistic cluttered forest simulation environment rendered using Microsoft AirSim (Shah et al., 2018a).

which cover a pre-defined area of the map (see Figure 4), preventing invalid initial locations such as inside the ground or trees or on the target vehicle, while covering the desired task area. The human demonstrator collected 90 demonstration trajectories in which the drone navigated to the fixed-location target. In each demonstration trajectory, the quadrotor's starting location was sampled randomly from the set of 24 possible locations, and the quadrotor's initial heading was also randomly sampled between 0 and 360 degrees. During the demonstrations, we recorded RGB images (224x224x3) and eye gaze coordinates. We address observed imbalances in the dataset as described in Appendix B.

We evaluate each learned policy by rolling it out in the environment. To evaluate each policy, we use a set of 10 quadrotor starting points; these are distinct from the 24 starting locations used during training and are depicted in Figure 4 in magenta. For each evaluation rollout starting position, the quadrotor's initial heading is sampled randomly. For each evaluated model, we perform 5 evaluation rollouts at each of the 10 evaluation start points, for a total of $5 \times 10 = 50$ evaluation rollouts. Using the same set of evaluation starting locations across all models helps to ensure evaluation consistency across approaches. Note that the randomly sampled heading is held constant for the 5 rollouts in each location, though it varies between locations.

## 4.3. Evaluation Metrics

We evaluate task performance in the evaluation rollouts via the following metrics (Anderson et al., 2018):

1. Task completion rate (TCR): this is the percent of episodes that are successful. In the stationary target task, a rollout is successful if the quadrotor navigates within 5 meters of the target vehicle's center of mass.

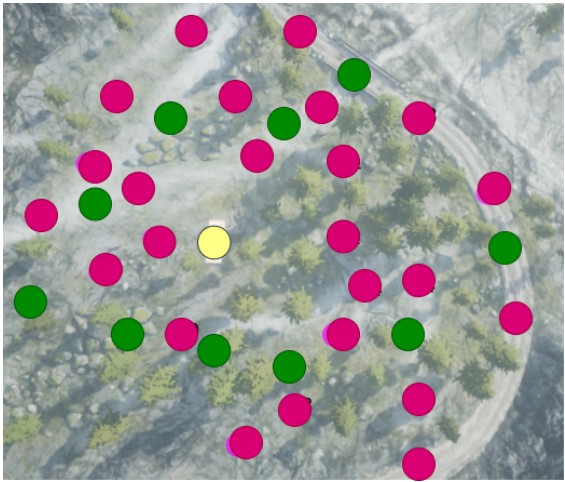

*Figure 4.* Bird's-eye view of the target vehicle location and drone start locations. The magenta points are the drone start locations in the demonstration dataset, the green points are the drone start locations during evaluation, and the single yellow point is the fixed location of the target vehicle in the stationary target task.

In the moving target task, an episode is similarly successful if the quadrotor navigates within 5 meters of the in-motion truck's center of mass.

2. Collision rate (CR): the percent of episodes in which the agent collides with the ground or any obstacles in the environment such as rocks and trees. The current episode is terminated when a collision occurs.

### 4.4. Comparison Methods

We compare GRIL with three imitation learning baseline methods: vanilla behavioral cloning (BC), Attention-Guided Imitation Learning (AGIL) (Zhang et al., 2018), and behavior cloning with a context-aware gaze loss (BC+CGL) (Saran et al., 2020). Appendix D specifies network architecture details for all baseline comparisons, while Appendix C specifies hyperparameter values for all methods, as well as hyperparameter ranges tested.

The BC baseline is trained to predict control commands directly from input images without gaze, whereas GRIL jointly predicts control commands and gaze coordinates. Note that the BC and GRIL network architectures are analogous, except that GRIL has a gaze regularization head that is not present in the BC model.

In AGIL, a gaze prediction network is trained to estimate a gaze heatmap, which is then inputted to the policy network. In contrast, GRIL leverages gaze prediction as a regularizer rather than a policy network input. Since AGIL was designed for discrete control of a 2D Atari game, we developed a custom variant of AGIL for performing continuous quadrotor control. Our implementation of AGIL is adapted

from the authors' implementation; we change the network architecture to process our dataset and its associated outputs and change the loss function from a classification loss to a regression loss.

The CGL baseline calculates an auxiliary loss that penalizes the policy network's final convolution layer for having low activations where the human's gaze is focused. Our auxiliary loss predicts gaze rather than penalizing network weights. As mentioned below and in the discussion, GRIL seems to require significantly less tuning than CGL. As with AGIL, we adapt the authors' implementation to our task.

### 4.5. Results

**Algorithm Performance.** Table 1 displays the performance of GRIL and all baseline comparisons. We see that GRIL yields the strongest performance, achieving the highest task completion rates (TCR) and lowest collision rates (CR) compared to the baseline methods. Meanwhile, Figure 6 provides an example qualitative comparison between the methods by depicting the evaluation rollout trajectories corresponding to a particular starting location in the stationary task. At this evaluation location, GRIL and BC-CGL successfully navigate to the target location, while BC mostly crashes with the terrain early and AGIL drifts away.

**Generalizing GRIL to Following a Moving Target.** We next evaluate how well GRIL generalizes to following a moving target when only trained with stationary target demonstrations. In this moving target task, we evaluate the same policies that were trained to search for and navigate toward a stationary target; the dataset in Section 4.2 does not include any demonstrations in which the target moved. This task is more difficult than the stationary target task, both because the policy must generalize to a task on which it was not trained and because the policy must track and follow the moving target.

Table 1 shows the performance of GRIL and comparisons on the moving target task. During these evaluation rollouts, GRIL, BC, and BC-CGL demonstrated a remarkable capability to generalize to the new task despite not having been previously trained on it, often successfully searching for and following the moving target. We see that GRIL outperforms each of the baseline methods in the moving target task, indicating that GRIL is able to generalize to novel tasks more robustly than previous state-of-the-art approaches. Qualitatively, we noticed that GRIL sometimes lost track of the target if the vehicle left visual sight; however, we also noticed that GRIL demonstrated recovery behaviors if the target re-appeared in sight. We provide videos of GRIL and baseline methods performing the moving target task on our supplemental website, `https://sites.google.com/view/gaze-regularized-il/`.

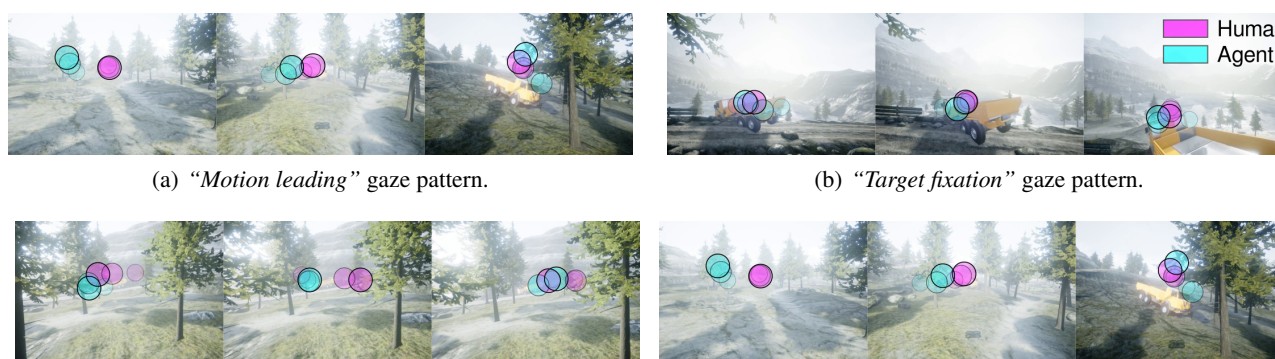

(a) *"Motion leading"* gaze pattern.

(b) *"Target fixation"* gaze pattern.

(c) *"Saccade"* gaze pattern.

(d) *"Obstacle fixation"* gaze pattern.

*Figure 5.* Visualization of a sequence of three frames illustrating characteristic gaze patterns observed in the human demonstrations, showed as magenta circles in the figure, which were also learned by the proposed model, showed as cyan circles (best viewed in color). The larger, less transparent circle illustrates the current gaze observation and the smaller, more transparent circles represent gaze (and gaze predictions) from previous timesteps.

|  | GRIL | | BC-CGL | | BC | | AGIL | |
|---|---|---|---|---|---|---|---|---|
|  | TCR (%) | CR (%) | TCR (%) | CR (%) | TCR (%) | CR (%) | TCR (%) | CR (%) |
| Stationary | **80** (±**9.9**) | **20** (±**9.9**) | 64 (±14.8) | 36 (±14.8) | 40 (±13.7) | 58 (±13.5) | 10 (±10.0) | 60 (±16.3) |
| Moving | **40** (±**12.3**) | **40** (±**9.4**) | 36 (±14.8) | 42 (±15.9) | 30 (±12.7) | 66 (±12.3) | 14 (±10.3) | 86 (±10.3) |

*Table 1.* Performance of GRIL and baselines in the stationary and moving target tasks. Notably, the moving target task evaluates the models' ability to transfer to a new task, since the demonstration data only included a stationary target. For both tasks, we compare the performance of GRIL, BC, AGIL (Zhang et al., 2018), and BC-CGL (Saran et al., 2020). We show the collision rate (CR) and task completion rate (TCR); values are mean (± standard error) over 10 starting locations, with 5 rollouts per starting location each in the static and moving target tasks.

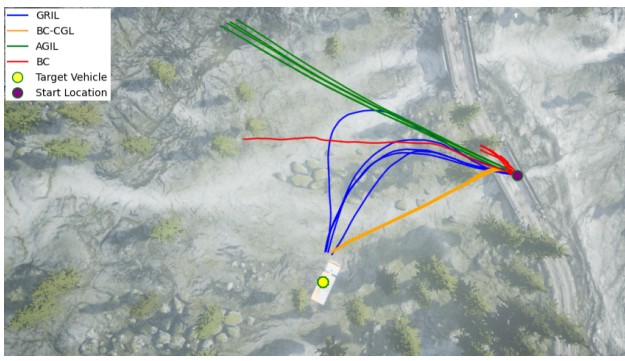

*Figure 6.* Evaluation rollout trajectories for GRIL and baseline comparisons in the stationary task. For one of the evaluation starting locations, we depict five evaluation rollouts corresponding to each method. The task consists of navigating from the starting location (purple dot) to the target vehicle (yellow dot).

**Gaze prediction results.** With respect to GRIL's gaze prediction performance, we observe that several distinct gaze patterns present in the human demonstrations were also learned by GRIL's gaze prediction head, as seen in Figure 5: a) a *"motion leading"* gaze pattern where gaze attends to the sides of the images followed by a yaw motion in the

same direction, as illustrated in Figure 5(a). This pattern is mostly observed at the beginning of the episode when the target vehicle is not in the agent's field-of-view; b) a *"target fixation"* gaze pattern where gaze is fixated on the target during the final approach, as illustrated in Figure 5(b). In this pattern, gaze is fixed at the top of the target vehicle, independent of the agent's current motion; c) a *"saccade"* gaze pattern where gaze rapidly switches between fixation on nearby obstacles (Salvucci & Goldberg, 2000), as illustrated in Figure 5(c). This pattern is characteristic when there are multiple obstacles between the current agent location and the target vehicle; and d) an *"obstacle fixation"* gaze pattern where the gaze attends to nearby obstacles when navigating to the target, as illustrated in Figure 5(d). This pattern is mostly observed when the quadrotor is close to a potential obstacle. This illustrates how GRIL's gaze prediction head is able to capture and replicate similar visual attention cues demonstrated by the user when performing the task.

## 5. Discussion

We propose the GRIL algorithm for leveraging human eye gaze in a context-aware imitation learning framework. GRIL leverages gaze via a novel, multi-objective optimization ap-

proach that jointly predicts control commands and gaze. The gaze prediction loss helps to regularize the policy network, so that it focuses on relevant parts of the image observations. This novel use of eye gaze makes the model sample efficient, practical, and generalizable. Our experiments demonstrate GRIL's performance in a continuous control setting featuring a quadrotor in an outdoor environment, in which GRIL outperforms state-of-the-art baseline methods AGIL and CGL. Though our experiments train the model to perform search and navigation toward a single stationary target, we show that GRIL can also perform the task with a moving target, and furthermore, that GRIL generalizes to the new task more robustly than baseline comparisons.

We speculate that, because GRIL is optimizing multiple objectives with action and gaze prediction heads, it may enable more robust and efficient learning. The benefit behind using a multi-objective optimization framework is twofold. Firstly, if a task is noisy or data is limited and high-dimensional, it can be difficult for a model to differentiate between relevant and irrelevant features. Multi-objective learning helps the model to focus its attention on the features that most matter, since optimizing multiple objectives simultaneously forces the model to learn relevant features that are invariant across each objective (Caruana, 1993; Abu-Mostafa, 1990; Ruder, 2017). Finally, a multi-objective learning framework, as proposed by this work, enables eye gaze prediction loss to act as a regularizer by introducing an inductive bias. As such, it reduces the risk of overfitting as well as the Rademacher complexity of the model, i.e. its ability to fit random noise (Baxter, 2000).

We hypothesize that GRIL may outperform CGL in part because it incorporates gaze in a more direct manner requiring fewer hyperparameters. While GRIL utilizes the gaze coordinate directly, CGL requires a gaze heatmap, for which one must specify a resolution and degree of spread about the gaze coordinates. CGL's heatmap also results in a more complex loss function leveraging the KL divergence, while GRIL relies on a simpler MSE loss. We also note that CGL imposes no additional network parameters to fit, whereas GRIL imposes an additional network head as it simultaneously predicts both control commands and gaze coordinates.

In regards to why GRIL significantly outperforms AGIL, we hypothesize that AGIL requires significantly more demonstration training samples compared to GRIL due to its separate gaze prediction network. AGIL requires first training a gaze prediction network to model human foveation by learning a gaze prediction heatmap, and then uses the gaze predictions to train a separate policy model. In contrast, GRIL combines both the gaze prediction network and policy network in a single model with branching network heads and shared weights. This significantly reduces the number

of parameters to learn.

**Future work.** Future work will include evaluating GRIL in additional environments, as well as to study the use of eye gaze in physical robot experiments. We are also excited to leverage eye gaze regularization in other problem settings, for instance multi-task learning and reinforcement learning. Another interesting avenue for future work is the addition of more human input modalities to the proposed approach, for instance natural language, to condition the model to perform multiple diverse tasks. The ability to use eye gaze data to leverage human visual attention opens the door to adapting this research to learning unified policies that can generalize across multiple human input types, task contexts, and task objectives.

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

# A. Details of the gaze data collection

### A.1. Eye Tracker Calibration

Before an eye tracking recording is started, the user is taken through a calibration procedure. During this procedure, the eye tracker measures characteristics of the user's eyes and uses them together with an internal, anatomically-correct 3D model of the human eye to calculate the requisite gaze data. This model includes information about shapes, light refraction, and reflection properties of the different parts of the eyes (e.g. cornea, placement of the fovea, etc.). During the calibration, the user is asked to look at specific points on the screen. These specific dots are also known as calibration dots. During this period several images of the human operator's eyes are collected and analyzed. The resulting information is then taken into account along with the eye model and the gaze coordinates (abcissa and ordinate in the screen frame) for each image.

### A.2. Gaze Data Collection

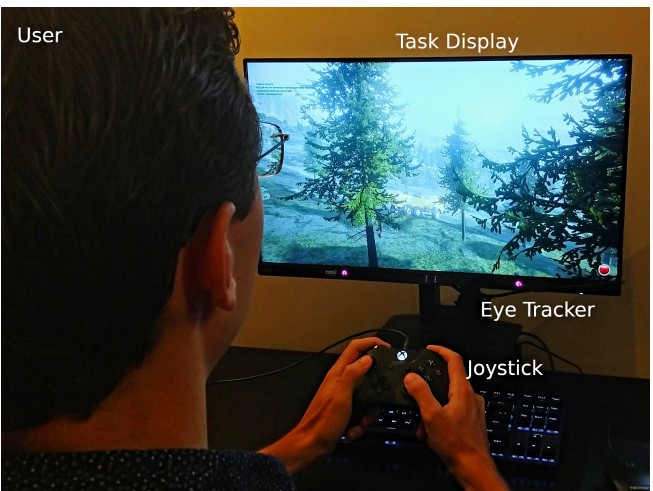

*Figure 7.* Data collection setup used to record human gaze and joystick data illustrating the relative positioning between the user, task display, joystick, and eye tracker. The user is given only the first-person quadrotor view while performing the visual navigation task.

The task was presented to the demonstrator on a 23.8 inches display with 1920x1080 pixel resolution, and eye gaze data collection was conducted using a screen-mounted eye tracker positioned at a distance of 61cm from the demonstrator's eye, as seen in Figure 7. Before data collection, the height of the demonstrator's chair was adjusted in order to position their head at the optimal location for gaze tracking. The eye tracker sensor was calibrated according to an 8-point manufacturer-provided software calibration procedure. The demonstrator avoided moving the chair and minimized torso movements during data collection while moving the eye naturally. The demonstrator was also given time to acclimate to the task until they judged for themselves that they were confident in performing it. The dataset was collected by a single human demonstrator.

# B. Addressing Data Imbalance

**Data imbalance.** The size of the dataset is limited due to human participation. During exploratory data analysis on the collected dataset, it was found that the yaw-axis actions were mostly centered around zero (see Figure 8), such that the agent has many more examples of flying straight than of turning.

**Data augmentation and undersampling.** We performed data augmentations to address the dataset imbalance. First, we created a horizontally-flipped version of each demonstration trajectory. The images were horizontally flipped with corresponding changes in the horizontal gaze coordinate and control commands (i.e., roll and yaw are reversed, while pitch and throttle are unchanged). Then we performed random undersampling of states with zero yaw, in which we rejected 90% of the zero yaw data samples. Figure 8 shows the distribution of control commands before and after the data augmentations were applied.

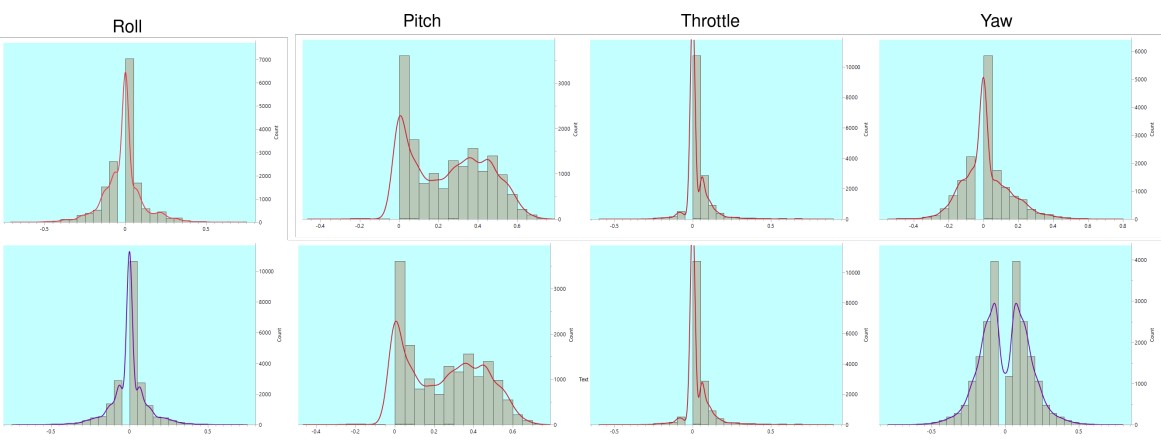

*Figure 8.* The distribution of control commands reflects the forward motion bias of the dataset. Most of the states have yaw commands centered around zero. This creates an imbalance in the dataset, which may result in a visuomotor policy where the vehicle learns to fly only straight.

**Weighted BC loss function.** For GRIL and all baseline comparisons, we predict actions by weighting the mean squared error BC loss $\mathcal{L}_{BC}(\theta)$ given by Equation (2). To help counter the forward motion bias in the dataset, the yaw control command was weighted more highly than the other three control parameters.

$$\mathcal{L}_{BC}^{\text{weighted}}(\theta) = \frac{1}{M} \sum_{i=1}^{M} \|w \odot (\pi_{\text{action}}(o_i \mid \theta) - a_i)\|^2,$$

which is analogous to the BC loss in (2) except that $w$ is a vector in which the $i^{\text{th}}$ element weights the $i^{\text{th}}$ control command component, and $\odot$ denotes an element-wise product. Recall that $M$ is the number of experience tuples in the demonstration dataset.

We manually tuned the weighting parameters $w$, and use a weight of 0.7 for the yaw command and weights of 0.1 for the other control components.

## C. Hyperparameters

We keep the following hyperparameters constant across all methods. The loss function is optimized using the Adam (Kingma & Ba, 2014) optimizer with a learning rate of 0.0003 for 30 epochs and batch size of 32 data samples. For training the model we have used an exponential decay function to schedule the decrease in learning rate. The decay occurs every 10,000 steps during model training.

For CGL, the loss function is a weighted sum of a BC term and the CGL term:

$$\mathcal{L}_{CGL}(\theta) = \lambda_1 * \mathcal{L}_{BC}(\theta) + \lambda_2 * \mathcal{L}_{CGL}(\theta),$$

where $\lambda_1$ and $\lambda_2$ are hyperparameters that determine the relative weighting between the terms. We tune $\lambda_1$ and $\lambda_2$ manually, and the optimized model uses $\lambda_1 = 0.9$ and $\lambda_2 = 0.1$. GRIL has analogous $\lambda_1$ and $\lambda_2$ hyperparameters that determine the weighting between the BC and gaze-regularization losses.

### C.1. BC-CGL

We tuned the following two gaze hyperparameters for BC-CGL: the kernel size of the gaze heatmap and the loss weighting. For the kernel size, the parameters considered were 3, 5, 10, and 15. For the loss weighting (action weight, gaze weight), the weightings considered were (0.95, 0.05), (0.9, 0.1), (0.7, 0.3), and (0.5, 0.5). In the table, bolded values indicate the hyperparameter values found to be optimal.

| Gaze Heatmap Kernel Size | 3 | 5 | **10** | 15 |
|---|---|---|---|---|
| Loss Weighting (action weight, gaze weight) | (0.95, 0.05) | **(0.9, 0.1)** | (0.7, 0.3) | (0.5, 0.5) |

## C.2. AGIL

The gaze hyperparameter for AGIL was the kernel size of the gaze heatmap. For the kernel size, the parameters considered were 3, 5, 10, and 15. In the table, the bolded value indicates the hyperparameter value found to be optimal.

| Gaze Heatmap Kernel Size | 3 | 5 | **10** | 15 |
|---|---|---|---|---|

## C.3. GRIL

The gaze hyperparameter for GRIL was the loss weighting. For the loss weighting (action weight, gaze weight), the weightings considered were (0.95, 0.05), (0.9, 0.1), (0.7, 0.3), and (0.5, 0.5). In the table, the bolded entry indicates the hyperparameter combination found to be optimal.

| Loss Weighting (action weight, gaze weight) | (0.95, 0.05) | **(0.9, 0.1)** | (0.7, 0.3) | (0.5, 0.5) |
|---|---|---|---|---|

# D. Baseline Model Architectures

We present in Figures 9, 10, 11, and 12 the network architectures used for each baseline method: BC, Attention-Guided Imitation Learning (AGIL) (Zhang et al., 2018), and Behavior Cloning with a Context-aware Gaze Loss (BC+CGL) (Saran et al., 2020), respectively.

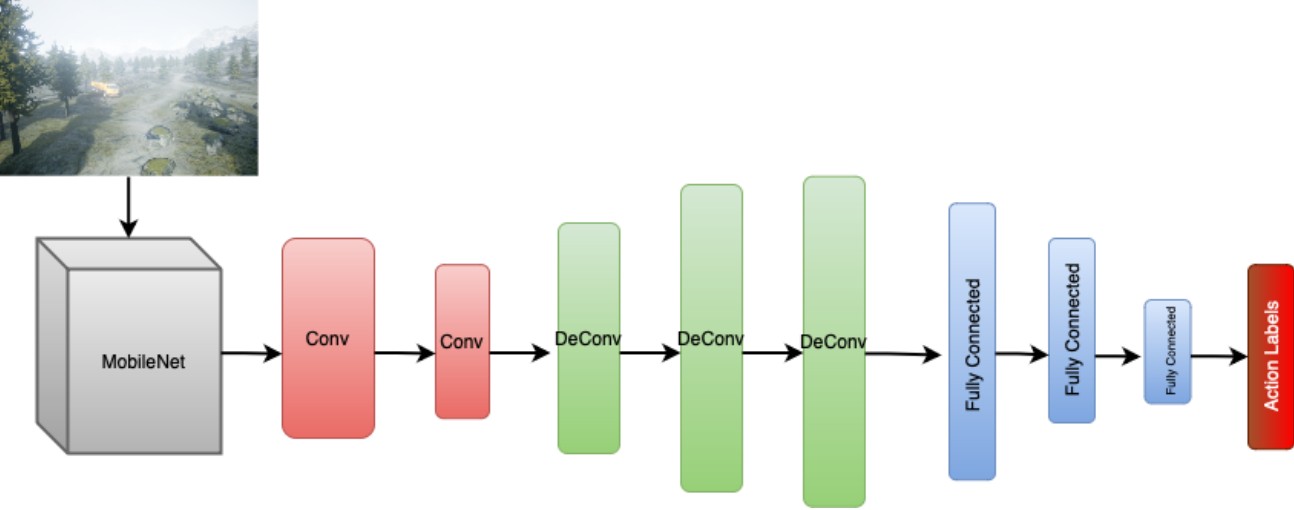

*Figure 9.* Standard behavior cloning model where only RGB images are used as input to the model.

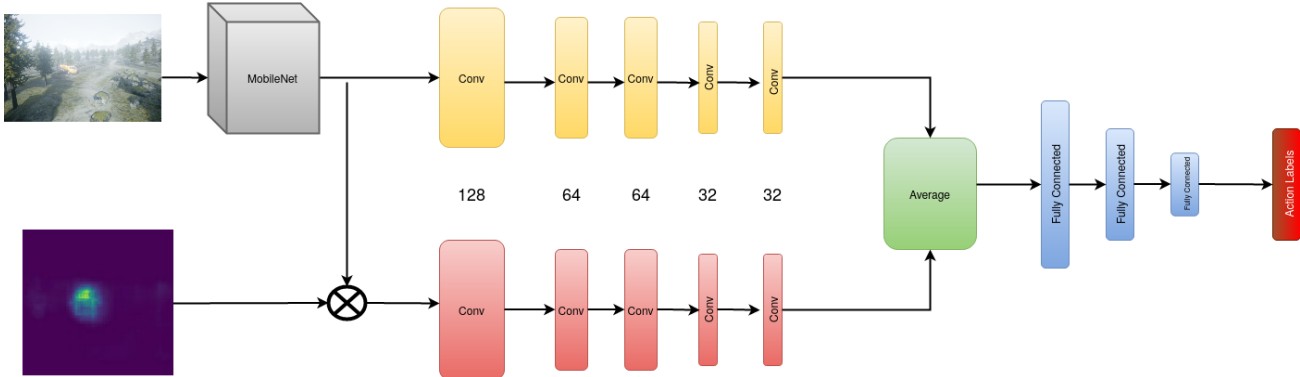

*Figure 10.* AGIL policy model architecture where RGB images and gaze heatmaps are used as an input to the model. The heatmap at the bottom-left is generated as shown in Figure 11.

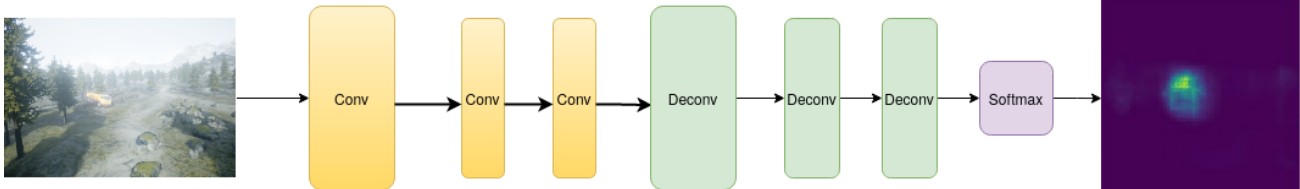

*Figure 11.* AGIL gaze prediction model architecture where RGB images are used as input and a predicted gaze heatmap is the output. This model produces the heatmap seen in Figure 10.

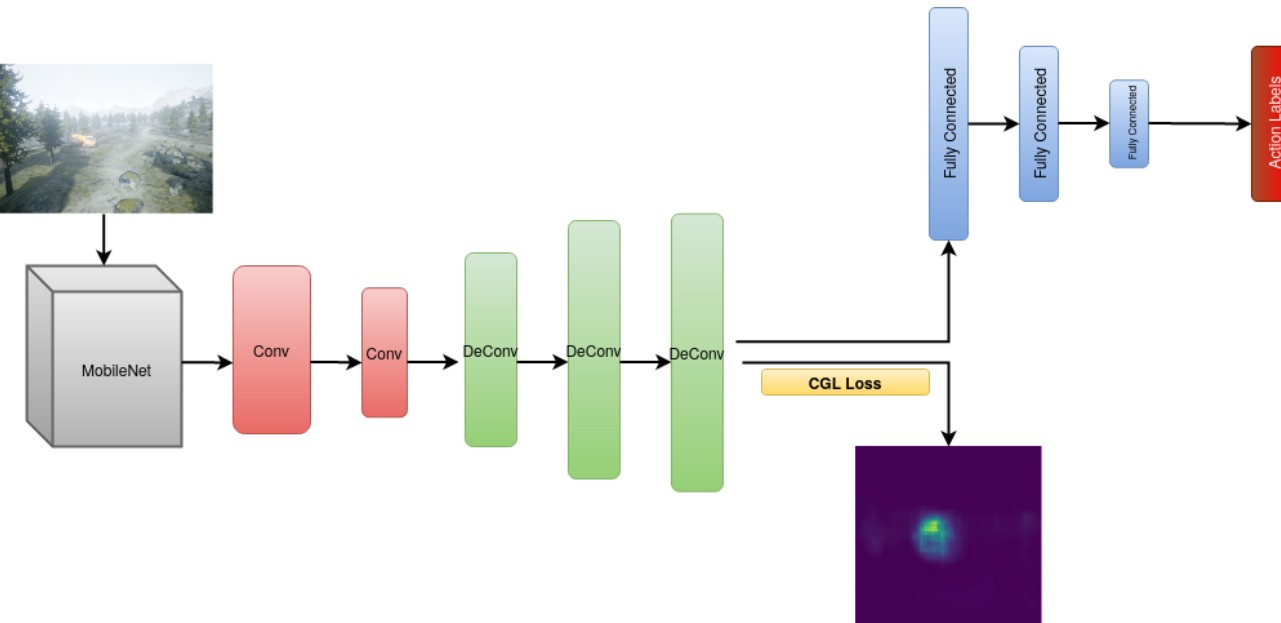

*Figure 12.* BC+CGL where RGB images are being used as an input to the model.