# OpenReview forum: "Imitation Learning with Human Eye Gaze via Multi-Objective Prediction"
_ICML.cc/2023/Workshop/ILHF — ILHF Workshop ICML 2023_

### Official Review · Reviewer_XR2R · 2023-06-16
**Interesting approach that uses eye gaze as training signal.**

**Rating:** 7
**Confidence:** 3

**Review:**

This paper proposed Gaze Regularized Imitation Learning (GRIL). In GRIL, in addition to human demonstrations, an agent learns from eye gaze to solve tasks where visual attention is helpful.


Quality: The paper is technically sound. The claims and proposed approach is supported by experimental results.

Clarity: The paper is generally well-written.

Originality and Significance: The reviewer found using gaze, a proxy of human attention, as training signal convincing. The proposed multi-objective architecture enables the approach to outperform competitive baselines.

---

### Official Review · Reviewer_RWGQ · 2023-06-17
**Worth Including in the Workshop**

**Rating:** 8
**Confidence:** 4

**Review:**

SUMMARY:
The authors present an end-to-end architecture for imitation learning in continuous control settings. They claim it is the first such architecture with a separate head for predicting human gaze. They demonstrate the functionality of their architecture in a complex drone navigation simulator and show that their system is also able to generalize to some unseen tasks in this simulator. They provide the dataset they train on for future researcher use.

COMMENTS:
If all the claims made by the authors are accurate, and to my knowledge, they are, then this work makes several meaningful and novel contributions to the field. However, their empirical results leave much to be desired. While their environment is complicated, the standard error in their results (if I'm reading this right) means they cannot reach strong conclusions using them. I believe this would pose a problem for this in appearing as a full conference publication, but for workshop publications, strong empirical results are not necessary. Apart from that issue, the paper itself is well thought out and clearly presented. I believe its release would have significant value to the community.

CONCLUSIONS:
I believe this paper is well above the bar for acceptance at this workshop. I believe it is absolutely within the scope of what would be relevant to this workshop.

---

### Decision · Program_Chairs · 2023-06-20

Accept